# Advancing Cross-Lingual Capabilities for Humanoid Robots: Leveraging Chinese NLP through Pictophonetic Advantages

## Abstract

Humanoid robots, as a critical trajectory in the development of artificial intelligence, are poised to play a key role in the era of cross-lingual and multimodal intelligence. This paper explores the unique capabilities of humanoid robots in multilingual processing by harnessing the pictophonetic advantages inherent in the Chinese language. Unlike phonetic languages such as English, Chinese characters encapsulate ideographic, phonetic, and semantic components within a single symbol, providing a rich, multidimensional data source. By analyzing the successful localization of the periodic table in Chinese, this study illustrates how the unique naming conventions used by Chinese chemists bridge scientific and linguistic understanding. It advocates adopting the systematic approach seen in Chinese chemical nomenclature to further advance research in Chinese natural language processing (CNLP). To this end, the Six-Writings Pictophonetic Coding (SWPC) technology is introduced, which constructs efficient character and word matrices to enable humanoid robots to process Chinese language inputs effectively. The integration of SWPC with techniques such as Scale-Invariant Feature Transform (SIFT) and machine learning facilitates multimodal recognition of characters and words, allowing robots to prioritize Chinese information and seamlessly process it in conjunction with other languages. This approach has the potential to significantly enhance natural language understanding and generation in complex Chinese contexts. By drawing insights from Chinese chemical nomenclature, the paper lays a foundation for intelligent cross-lingual interactions, providing a new direction for CNLP research and paving the way for humanoid robots to achieve deeper integration into future intelligent societies.

## 1 Introduction

Humanoid robots are emerging as pivotal agents in the development of artificial intelligence, playing an essential role in bridging human-machine interactions across languages and modalities (Saeedvand et al., 2019; Kuanysh, 2024). As AI progresses towards more advanced cross-lingual capabilities, leveraging the strengths of different languages becomes crucial.

Unlike phonetic languages such as English, Chinese characters encapsulate ideographic, phonetic, and semantic components within a single symbol, providing a rich, multidimensional data source. This makes Chinese characters a compact and information-rich medium for conveying complex concepts. Integrating Chinese characters into the language processing framework of humanoid robots can offer unique advantages, enhancing their ability to process and generate multilingual outputs with higher efficiency. By analyzing the successful localization of the periodic table in Chinese, this study illustrates how the unique naming conventions used by Chinese chemists bridge scientific and linguistic understanding. It advocates adopting the systematic approach seen in Chinese chemical nomenclature to further advance research in Chinese natural language processing (CNLP).

Figure 1 illustrates the example elements from the "Periodic Table of Chemical Elements" (hereinafter referred to as Periodic Table): platinum (Pt) in the English version and hydrogen (H) in the Chinese version. In a limited space, information such as 1) atomic number, 2) element symbol, 3) element name, and 4) atomic mass is presented. However, due to the unique nature of Chinese char-

acters, even readers with only a basic understanding of chemistry can infer additional information from the Chinese names of elements. For instance, the Chinese character for hydrogen (氢) suggests embedding information that it is a gas and that it is lightweight.

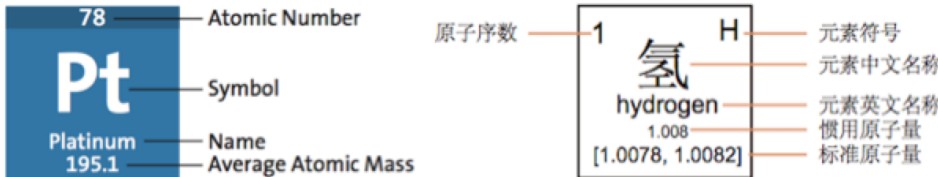

Figure 1: Example elements from the Periodic Table: platinum (Pt) in the English version and hydrogen (H) in the Chinese version, see Figure 8 in appendix.

In this context, we pose the following question: if a multilingual robot were to extract information from the Periodic Table or similar sources, should it prioritize reading the Chinese version over other languages? While existing NLP models have achieved significant success in processing phonetic languages like English, they often lack the specialized frameworks needed to harness the structural and semantic richness inherent in logographic languages such as Chinese. To address this gap, we introduce the Six-Writings Pictophonetic Coding (SWPC) approach (Weigang et al., 2024a). This approach aligns conceptually with the systematic representation of chemical elements in the Chinese periodic table, thereby enabling humanoid robots to optimize the processing of scientific and technical data. A detailed analysis will be presented in three areas, leading to some innovative approaches to Chinese NLP:

- **Historical Culture and Modern Technology:** We'll review the formation of the Chinese Periodic Table and compare the advantages of Chinese characters in expressing scientific terms. Chinese chemical nomenclature techniques will be summarized for application in humanoid robot language processing.

- **New Chinese Character and Word Encoding Framework:** The "Six-Writings Pictophonetic Coding (SWPC)" will be utilized to construct the Character Radical-Component Matrix (CRCM) for character generation, and the Lexical Affix-Root Matrix (LARM) for word generation. This framework enables pictophonetic and semantic recognition with dynamic generation capabilities, supported by dedicated multimodal processing algorithms for both images and text.

- **External Visual Language and Internal Processing Language:** Scale-Invariant Feature Transform (SIFT) and machine learning technologies will optimize robotic language processing. "Once Learning" method and character formation rules will enhance character and word recognition, improving humanoid robot language processing in multimodal scenarios.

By integrating phonetic and semantic language systems, we aim to develop a new Chinese language processing model that significantly improves humanoid robot natural language understanding and generation in complex Chinese environments. This will lay the foundation for cross-language intelligent interaction and promote the integration of humanoid robots into future intelligent societies.

## 2 KEY TECHNICAL CHARACTERISTICS OF HUMANOID ROBOTS

As vital components of future intelligent societies, humanoid robots' technical capabilities will significantly shape their real-world applications (Jiang et al., 2024). This section explores these unique abilities across several dimensions. Although this paper does not focus exclusively on humanoid robotics, understanding these aspects is crucial for integrating large language models (LLMs) in their development.

- **Multimodal Information Processing:** Humanoid robots replicate human anatomical features but surpass human sensory limits. Unlike humans, robots can achieve 360-degree perception, integrating visual, auditory, olfactory, and tactile inputs, including infrared and thermal data. This multimodal integration allows robots to operate effectively in dynamic

environments. For example, in crowded spaces, they can detect movement patterns and adjust their actions.

- **Superhuman Information Processing:** Humanoid robots can process data beyond human perception, such as infrared or ultraviolet light and inaudible sound waves. This capability is invaluable in critical environments like industrial sites or healthcare, where robots can detect radiation leaks or monitor subtle physiological signs. High-performance processors and advanced sensors enable these robots to excel in areas such as medical diagnostics and industrial safety monitoring.

- **Cross-Lingual Language Processing:** Large language models (LLMs) with multilingual abilities empower humanoid robots to switch effortlessly between languages (Vaswani et al., 2017; Touvron et al., 2023). This is crucial in multicultural environments. However, logographic languages like Chinese pose unique challenges compared to phonetic languages. Chinese NLP involves interpreting characters with multiple meanings and complex syntax. Advances in LLMs now offer the potential for robots to overcome these hurdles, using deep learning to better process and generate languages like Chinese.

- **Multimodal Integration:** Robots need to integrate sensory inputs such as vision, sound, and touch to perform complex tasks efficiently. This multimodal ability allows them to recognize faces, voices, objects, gestures, and emotions. For instance, a robot navigating a hospital might simultaneously understand speech, recognize patients, and use tactile feedback to avoid obstacles.

- **Individual and Collective Intelligence:** Humanoid robots exhibit advanced individual intelligence, but collective intelligence is becoming increasingly important with networked communication. Robots can share information, collaborate, and optimize their behavior through self-organization and learning, significantly enhancing task efficiency.

- **Addressing the Needs of Chinese Natural Language Processing (CNLP):** Chinese, as a logographic language, presents distinct challenges for NLP. Its character-based system and wide semantic range require more sophisticated context processing. Robots must go beyond basic syntax to comprehend the deeper semantics of Chinese characters, sentences, and contexts to effectively interact in Chinese-speaking environments.

Humanoid robots equipped with multimodal information processing capabilities can leverage SWPC's comprehensive character representation to enhance language understanding. By integrating phonetic, semantic, and visual features, robots can achieve more nuanced cross-lingual communication.

## 3 INTRODUCTION TO THE CHINESE VERSIONS OF THE PERIODIC TABLE

Humans are typically proficient in only one or a limited number of languages, which reflect their nationality, culture, and identity. In contrast, humanoid robots are not constrained by language barriers. Their multilingual processing capabilities surpass human cognitive limitations, driven by advancements in technology. Robots can rapidly acquire and utilize various languages, thus serving as a bridge across cultural and linguistic divides.

To optimize language resource processing and enhance robotic performance, we examine the Periodic Table and the nomenclature of organic compounds as illustrative examples (Wang, 2010). While humans generally read the Periodic Table in their native language, robots must decide which language version to prioritize. By analyzing these cases, we can gain valuable insights into improving robotic efficiency in complex multilingual environments.

### 3.1 ADVANTAGES OF THE CHINESE VERSION OF THE PERIODIC TABLE

In 1869, Russian chemist Dmitri Mendeleev revolutionized chemistry and atomic science with the periodic table. For phonetic languages like English and Portuguese, translating the periodic table was relatively straightforward. However, adapting it into Chinese presented significant challenges. Early Sino-cultural regions, such as Japan, relied on phonetic transliterations, leading to confusion due to inconsistent usage. Some elements were represented by single characters, while others required multiple characters.

The Chinese periodic table not only addresses these issues of localization but also leverages the unique advantages of Chinese characters, which encapsulate visual, phonetic, and semantic features. For instance, the element "At" is symbolized by "砹", where the "石" (stone) radical intuitively suggests a non-metal, while "艾" (Ai) provides the phonetic hint. Similarly, "Li" is denoted by "锂", where "钅" indicates metallic properties, and "里" contributes the phonetic element. Such phono-semantic combinations simplify the representation and understanding of complex chemical elements, making the Chinese version more efficient for both human learning and machine processing.

Furthermore, the standardized rules for creating new element characters support the generation of symbols for newly discovered elements. For example, on November 30, 2016, the International Union of Pure and Applied Chemistry (IUPAC) announced the English names and symbols for four new elements, completing the seventh period of the periodic table. On May 9, 2017, the Chinese Academy of Sciences, the National Language Committee, and the National Science and Technology Terminology Committee jointly released their Chinese names: "钅尔" (nǐ), "镆" (mò), "石田" (tián), and "气奥" (ào).

Figure 2 presents examples of four elements (fluorine, neon, chlorine, and argon) from the periodic tables of English, Chinese, Japanese, and Portuguese. In comparison, the English and Portuguese versions are relatively concise. The Chinese version, however, presents a wealth of information in a more concise and eye-catching manner within the same space. The Japanese version is more complex, with elements represented by either 1-2 kanji characters (e.g., "塩素" for chlorine), a combination of kanji and kana (e.g., "弗(フッ)素" for fluorine), or entirely phonetic kana (e.g., "アルゴン" for argon). Preliminary statistics indicate that English element names average 7.82 letters in length, which corresponds to 7.82 bytes in Unicode/UTF-8 encoding, with a maximum of 13 letters (and bytes). For Latin language, this average increases slightly to 8.07 letters and bytes, also with a maximum of 13. In contrast, Japanese element names average 4.58 characters in length, occupying approximately 13.73 bytes in Unicode/UTF-8 encoding, with a maximum of 9 characters, taking up to 27 bytes. Additionally, 55% of the element names exceed the median value of 5 characters. Meanwhile, each Chinese element name is represented by a single character, requiring only 3 bytes in Unicode/UTF-8 encoding—making it significantly more concise.

Figure 2: Examples of four elements (fluorine, neon, chlorine, and argon) from the periodic tables of English, Chinese, Japanese, and Portuguese, see Figure 8 in appendix.

Chinese scholars insisted on using single Chinese characters, even creating new ones, without resorting to phonetic annotations or pinyin. This practice demonstrates the foresight of Chinese scholars in promoting science popularization and linguistic innovation at that time. This attempt to seamlessly integrate scientific concepts with linguistic symbols remains highly valuable even in today's era of large language models.

Chinese element symbols go beyond being mere chemical representations; they embody a unique cultural phenomenon. By skillfully integrating the phonetic and semantic characteristics of Chinese characters with scientific concepts, they vividly illustrate the depth and versatility of the language. This method of translating abstract scientific ideas into tangible symbols not only aids in research and education but also adds new layers of meaning to Chinese culture. Due to the compactness of Chinese characters, information can be expressed with greater efficiency, reducing cognitive load and enhancing machine processing capabilities. In particular, the phonosemantic structure of Chinese characters optimizes encoding, making computational processing more efficient. This fusion of science and the humanities has created a valuable cultural legacy for future generations.

### 3.2 GENERATING COMPOUND NAMES BASED ON AFFIX AND ROOT CHARACTERS

The naming of Chinese chemical element symbols marks only the beginning of a great endeavor. On this foundation, the naming of organic and inorganic chemical compounds unfolds its vast potential (Wang, 2010). The Chinese Chemical Society introduced the "Nomenclature of Organic Compounds-2017," (Society, 2017) establishing principles such as systematization, standardization, simplification, and consistency, which have fostered the orderly and scientific development of chemical research and education (Ma et al., 2019).

The naming of organic compounds in Chinese is far more intricate than that of elements or inorganic compounds. Prior to the 20th century, organic compound names were phonetically transliterated from Western languages, leading to cumbersome terms (He, 2016). To resolve this, chemists introduced single-character names like "烷" (alkane), "烯" (alkene), and "炔" (alkyne) based on semantic translation. These concise terms convey the core meaning of the compounds and lay the groundwork for naming their derivatives. For example, the Chinese character for "烃" (hydrocarbon) integrates elements from "碳" (carbon) and "氢" (hydrogen), with a "火" radical symbolizing combustion. Hydrocarbons, made of carbon and hydrogen, are the basis for many organic compounds such as alkanes, alkenes, and alkynes. The IUPAC nomenclature system (Favre & Powell, 2014) is often applied, with single bonds indicating alkanes, double bonds for alkenes, and triple bonds for alkynes. Cyclic structures are named as cycloalkanes, and those with benzene rings as aromatic hydrocarbons (He, 2016). This alignment with chemical "semantics" demonstrates the evolution from element names to more complex compound terms.

In discussing word formation, prefixes, suffixes, connectors, and roots in Chinese compound words are crucial. The "Nomenclature of Organic Compounds-2017" provides detailed naming rules for organic compounds, which cannot be fully covered here. However, the use of Chinese numerals and ordinal terms like the Heavenly Stems gives an insight into this system. The terms like "伯" (primary), "仲" (secondary), "叔" (tertiary), and "季" (quaternary) denote the substitution level of hydrogen atoms in hydrides. For example, hydrocarbons have primary, secondary, tertiary, and quaternary carbons, while amines correspond to primary, secondary, tertiary amines, and quaternary ammonium salts. These terms parallel their English counterparts and align Chinese and English naming conventions (Society, 2017).

The Chinese character element symbols go beyond simple chemical notation and represent a cultural phenomenon, creatively merging phonetic and semantic elements with scientific knowledge. This method makes complex scientific ideas more tangible and supports research and education, while also enriching the cultural heritage of Chinese characters.

Moreover, applying Chinese characters in chemical nomenclature, especially in constructing compound words, offers valuable insights for CNLP. This is a key point of this article. The efficiency of Chinese characters in expressing information reduces computational complexity, making language models more effective. For cross-lingual robots, the high information density of Chinese characters offers unique advantages. When processing the periodic table or compound names, humanoid robots, with their advanced multilingual processing capabilities, are able to efficiently acquire and utilize multiple languages, serving as a bridge across cultural and linguistic barriers.

## 4 APPLICATION OF SIX-WRITINGS PICTOPHONETIC CODING (SWPC)

Just as Chinese characters integrate the features of ideographic, phonetic, and semantic components in the naming of chemical elements and organic compounds, the concept of character formation and word creation offers valuable insights for language representation in multimodal Chinese processing. However, the challenge lies in how to translate the phono-semantic features of characters into computer-readable code to efficiently generate characters and words, which is the focus of this section. Modern robots must integrate information from various sensory channels, such as vision, hearing, and touch, and the multidimensional characteristics of Chinese characters align well with this requirement. The core content of this section is how to identify a Chinese character image with the help of SWPC. By the way, this article will not discuss visual scan related technologies for the time being.

### 4.1 USING SWPC TO PRESENT CHINESE CHARACTER AND WORD

Many large language models use Unicode/UTF-8 encoding to represent different language symbols and achieve tokenization Wang et al. (2020). In this system, English letters are encoded using ASCII, with each letter occupying one byte, while Chinese characters typically occupy 3-4 bytes. This encoding system effectively addresses the problem of unified multilingual representation and performs well in handling out-of-vocabulary (OOV) words. However, the processing of Chinese characters still faces challenges in efficiency and in the adequate representation of linguistic features. Particularly in natural language processing tasks, accurately expressing the phono-semantic structure of Chinese characters remains a technical bottleneck that needs to be overcome.

In the past, researchers have attempted to use systems such as the Four-Corner Code, Wubi, and Cangjie to aid natural language processing tasks through phonetic and structural encoding (Cao et al., 2020; Wu et al., 2020; Sun et al., 2021; Jin et al., 2021; Lv et al., 2022). However, these methods often encounter representation threshold issues in both encoding and image processing applications (Weigang et al., 2024b). For instance, among the 8,105 of the list of Commonly Used Standard Chinese Characters (Commission, 2013), the Wubi encoding has a threshold of 14 strokes, covering 83.54% of the characters, while the Cangjie encoding has a threshold of 16 strokes, covering 92.39%. Against this backdrop, SWPC method was developed, providing a foundational tool for multimodal Chinese character processing (Weigang et al., 2024a).

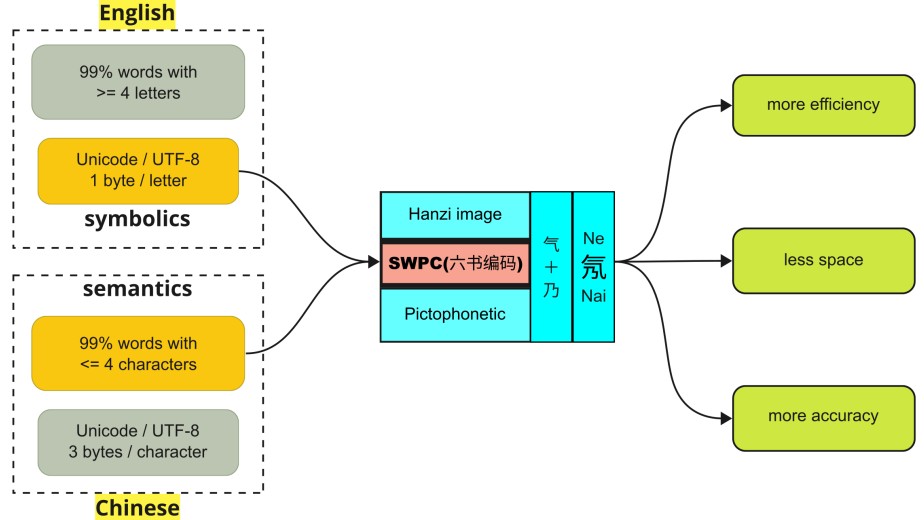

Figure 3: The concept of the Six-Writings pictophonetic coding (SWPC) method.

Unlike fixed encoding schemes such as Wubi or Cangjie, which encounter difficulties when handling characters that exceed a certain complexity threshold, SWPC employs a dynamic encoding strategy. This flexibility enables SWPC to represent a broader range of Chinese characters, making it more suitable for multimodal and cross-lingual applications (Weigang et al., 2024a). The SWPC technology encodes Chinese character radicals and components using two-letter combinations, resulting in approximately 676 possibilities (26 x 26 letters). Based on this foundation, the SWPC code for most Chinese characters typically consists of 4 letters, while more complex characters may require 6 to 12 letters.

Although English letters are represented by a single byte in the Unicode/UTF-8 system, 99% of English words are composed of at least four letters. The SWPC model leverages this by using English letters as symbolic representations of Chinese radicals and components, taking advantage of the fact that one letter corresponds to one byte in Unicode/UTF-8 encoding. At the same time, it preserves the grammatical rules of Chinese (semantic properties), which is beneficial since 99% of Chinese words consist of no more than four characters. This combination of symbolic and semantic encoding is the core of the SWPC approach, see Figure 3.

Table 1: A comprehensive comparison of the expressions of compounds in various languages

| Item | H | $^1H$ | $^2H$ | $^3H$ |
|---|---|---|---|---|
| English | Hydrogen | Protium | Deuterium | Tritium |
| Portuguese | Hidrogênio | Protium | Deutério | Trítio |
| Chinese | 氢(qīng) | 氕(piē) | 氘(dāo) | 氚(chuān) |
| Unicode | U+6C22 | U+6C15 | U+6C18 | U+6C1A |
| Kanji | 水素 | 軽水素 | 重水素 | 三重水素 |
| Kana | ソフトウェア | プロチウム | デュテリウム | トリチウム |
| SWPC | Rncf | Rntr | Rnjl | Rnkd |

Table 1 presents a detailed comparison of compound expressions across multiple languages, including English, Portuguese, Chinese, Unicode, SWPC, and Japanese Kanji/Kana. Based on this analysis, the following conclusions can be drawn:

- **Representation of Hydrogen Isotopes:** The table lists the three stable isotopes of hydrogen—protium, deuterium, and tritium—using distinct Chinese characters that correspond to their increasing atomic masses. Protium (atomic mass 1) is represented by "氕" (piē), which is formed with a single-stroke component and the radical "气" (qì, meaning 'gas'). Deuterium (atomic mass 2) is represented by "氘" (dāo), composed of a two-stroke component and the radical "气". Tritium (atomic mass 3) is represented by "氚" (chuān), which consists of a three-stroke component and the radical "气". These characters reflect a clear visual and semantic progression as the atomic mass increases, illustrating the integration of Chinese characters' semantic, phonetic, and graphic elements.

- **Compound Name Generation:** Once the basic naming of an element or compound is established, the names of related compounds can be generated systematically. For instance, "氘代甲醇" (deuterated methanol), a compound formed by the characters "氘" (dāo), "代" (dài), "甲" (jiǎ), and "醇" (chún), requires 16 letters when encoded using SWPC. By comparison, its English equivalent "Deuterated methanol" uses 19 letters, while its Japanese equivalent "重水素化メタノル" consists of 9 Japanese characters.

- **Encoding Efficiency Analysis:** When translated into Unicode/UTF-8 encoding, "氘代甲醇" requires 12 bytes in Chinese, "Deuterated methanol" requires 19 bytes in English, and "重水素化メタノル" requires 27 bytes in Japanese. By contrast, the SWPC representation of "氘代甲醇" uses 16 bytes and concurrently encodes both the semantic and phonetic features of the Chinese characters, highlighting its compact and multimodal nature.

## 4.2 MATRICES FOR CHINESE CHARACTER AND WORD GENERATION

To better understand SWPC application, we introduce the Character Radical-Component Matrix (CRCM), see Figure 4. In CRCM, the horizontal axis lists Chinese character radicals and their corresponding SWPC codes in red, such as "气 - Rn" (Qi), "石 - Do" (Shi), and "钅 - Qf" (Jin). The vertical axis lists character components and their SWPC codes in blue, such as "亚 - sl" (Ya), "同 - mz" (Tong), and "申 - jh" (Shen).

By combining radicals and components from the matrix, relevant Chinese characters are generated along with their SWPC codes, for example, "氩 - Rnsl" (Argon), "铜 - Qfmz" (Copper), "砷 - Dojh" (Arsenic), and "氯 - Rnvi" (Chlorine), among others.

Chinese vocabulary formation primarily relies on combining word roots according to grammatical relationships(Ruomei, 2014). This is exemplified in organic compound naming, which flexibly integrates Chinese word formation methods, chemical element naming rules, and IUPAC standards. To apply these insights to Chinese natural language processing, we introduce the Chinese Lexical Affix-Root Matrix (LARM). As shown in Figure 5, LARM is a valuable tool for understanding Chinese word formation.

For example, in LARM, the horizontal axis lists prefixes in red, such as "甲 (first)" with SWPC code "Ly" and "乙 (second)" wit SWPC code "Np". The vertical axis lists word roots in blue, such

| | Components/SWPC/Unicode（部件） | | | | |
|---|---|---|---|---|---|
| R a d i c a l 部首 | | 亚 (sl) \u4e9a | 同 (mz) \u540c | 申 (jh) \u7533 | 录 (vi) \u7533 |
| | 气(Rn) \u6c14 | 氩(Rnsl) Argon \u6c29 | | 氟(Rnjh) \u6c20 | 氯(Rnvi) Chlorine \u6c2f |
| | 石(Do) \u77f3 | | 硐(Domz) \u7850 | 砷(Dojh) Arsenic \u7837 | 碌(Dovi) \u788c |
| | 钅(Qf) \u9485 | 钯(Qfsl) \u94d4 | 铜(Qfmz) Copper \u94dc | | |

Figure 4: Chinese Character Radical-Component Matrix (CRCM) for some chemical elements.

as "醇 (alcohol)" with SWPC code "Sloc" and "酸 (acid)" with SWPC code "Slcr". By combining prefixes and word roots, LARM can generate new Chinese words. For instance: "甲醇 (methanol)" is represented by "Ly Sloc", and "乙酸 (ethanoic acid)" is represented by "Np Slcr".

| | Chemical root character | /SWPC/Unicode（化学根词） | | |
|---|---|---|---|---|
| P r e f i x 词缀 | | 醇(Sloc) U+9187 | 酸(Slcr) U+9178 | 烷(Oopffd) U+70F7 | 炔(Oonw) U+7094 |
| | 甲(Ly) U+7532 | 甲醇 Methanol | 甲酸 Formic acid | 甲烷 Methane | 甲炔 Methylidyne |
| | 乙(Np) U+4E59 | 乙醇 Etanol | 乙酸 Ethanoic acid | 乙烷 Ethane | 乙炔 Ethyne |
| | 丙(Gr) U+4E19 | 丙醇 Propanol | 丙酸 Propanoic acid | 丙烷 Propane | 丙炔 Propyne |

Figure 5: Chinese Lexical Affix-Root Matrix (LARM) for some organic compounds.

The proposed character generation matrix and word generation matrix realize the encoding generation of SWPC, laying a solid foundation for the next step in Chinese character image recognition.

### 4.3 CHARACTER IMAGE/TEXT PROCESSING FOR ENHANCED CHARACTER AND WORD RECOGNITION

This research integrates SWPC with advanced image processing techniques to develop multimodal algorithms for character and word recognition. We have constructed an SWPC database containing 3,981 Chinese characters (Weigang et al., 2024a) and an image library featuring 33,950 distinct character representations (Weigang et al., 2024b). The Scale-Invariant Feature Transform (SIFT) technology (Lowe, 1999) is applied to optimize these algorithms, allowing for effective recognition of Chinese characters and words in various contexts (Hu et al., 2013). While some state-of-the-art (SOTA) deep learning algorithms, like MA-CRNN (Tong et al., 2020) and MaskOCR (Lyu et al., 2024), can achieve higher accuracy in certain scenarios, SIFT's interpretability and compatibility with SWPC make it an ideal choice for providing a comprehensive understanding of the processing and recognition of Chinese characters and words.

Figure 6 illustrates how SIFT recognizes Chinese character images representing various chemical elements. Each image has a resolution of 96x96 pixels, and SIFT analyzes the radicals and components to identify the entire character. The recognition process involves the following steps:

- **Taking the character image by Once Learning:** The entire Chinese character image will be input into the system at once for subsequent processing (Weigang & da Silva, 1999).
- **Radical Recognition:** Some radicals are standalone Chinese characters that serve as the foundational elements for image recognition. These radicals have been pre-labeled with their corresponding SWPC codes. For example, the red characters shown in the left rows of CRCM in Figure 4 include "气 Rn" (Air), "石 Do" (Stone), and "钅 Qf" (Metal).

- **Component Recognition:** Some components also exist as independent Chinese characters and are used as foundational component images, each labeled with its corresponding SWPC codes. The blue characters in the top columns of CRCM in Figure 4 include "亚 sl" (Asia), "同 mz" (Same), "申 jh" (Apply), and "录 vi" (Record).

- **Character Identification:** Using the SIFT method, the system identifies each Chinese character by matching its radicals and components within the matrix structure of Figure 4. Once the character's radicals and components are detected, the SWPC code is generated. For instance, the following SWPC codes are generated for the characters: "氩 - Rnsl" (Argon), "铜 - Qfmz" (Copper), "砷 - Dojh" (Arsenic), and "氯 - Rnvi" (Chlorine).

- **Database Lookup:** The generated SWPC code is then used to retrieve the corresponding Chinese character (using its Unicode encoding) from the database, facilitating accurate recognition and representation of the character.

| 96x96 | Components/SWPC/Unicode（部件） | | | |
|---|---|---|---|---|
| | 亚(sl) \u4e9a | 同(mz) \u540c | 申(jh) \u7533 | 录(vi) \u5f55 |
| 气(Rn) \u6c14 | 氩(Rnsl) Argon \u6c29 0.2285 0.1896 | | 氮(Rnjh) \u6c20 0.2183 0.1698 | 氯(Rnvi) Chlorine \u6c2f 0.2062 0.1483 |
| 石(Do) \u77f3 | | 硐(Domz) \u7850 0.0384 0.1878 | 砷(Dojh) Arsenic \u7837 0.0384 0.2528 | 碌(Dovi) \u788c 0.0337 0.2326 |
| 钅(Qf) \u9485 | 钲(Qfsl) Softsteel \u94d4 0.1700 0.2094 | 铜(Qfmz) Copper \u94dc 0.1639 0.1661 | | |

Figure 6: Using SIFT to recognize Chinese Character Radical-Component Matrix (CRCM) for some chemical elements.

Figure7 demonstrates how SIFT is applied to recognize Chinese character images for various compound names. Each compound name image has a resolution of 48x48 pixels, and the SIFT algorithm identifies the entire name by analyzing its prefix and root characters. The recognition process involves the following steps:

| 48x48 | Chemical root character/SWPC/Unicode（化学根词） | | | |
|---|---|---|---|---|
| | 醇(Sloc) U+9187 | 酸(Slcr) U+9178 | 烷(Oopffd) U+70F7 | 炔(Oonw) U+7094 |
| 甲(Ly) U+7532 | 甲醇 Methanol 0.3143 0.3012 | 甲酸 Formic acid 0.3146 0.2688 | 甲烷 Methane 0.3143 0.2911 | 甲炔 Methylidyne 0.3273 0.2981 |
| 乙(Np) U+4E59 | 乙醇 Etanol 0.3204 0.3012 | 乙酸 Ethanoic acid 0.3204 0.2996 | 乙烷 Ethane 0.3329 0.2984 | 乙炔 Ethyne 0.3272 0.3029 |
| 丙(Gr) U+4E19 | 丙醇 Propanol 0.3081 0.3012 | 丙酸 Propanoic acid 0.3308 0.2688 | 丙烷 Propane 0.3081 0.2911 | 丙炔 Propyne 0.3081 0.3229 |

Figure 7: Using SIFT to recognize Chinese Lexical Affix-Root Matrix (LARM) for some organic compounds.

- **Taking the word image by Once Learning:** The entire Chinese word image will be input into the system at once for subsequent processing (Weigang & da Silva, 1999).

- **Prefix Character Recognition:** Certain prefix characters are used as the foundational prefix images, pre-labeled with their corresponding SWPC codes. For example, in the left rows of LARM in Figure 5, the SWPC code for "甲" (first) is Ly, and for "乙" (second), it is Np.

- **Root Character Recognition:** Specific root characters are used as foundational root images, each labeled with its corresponding SWPC codes. In the top columns of LARM in Figure 5, the SWPC code for "醇" (Alcohol) is Sloc, and for "酸" (Acid), it is Slcr, among others.

- **Compound Name Identification:** The SIFT algorithm identifies compound names by recognizing and matching the prefix and root characters within the matrix shown in Figure 5. After detecting the characters, their SWPC codes are generated. For example, "甲醇" (Methanol) has the SWPC code Ly Sloc, while "乙酸" (Ethanoic acid) has the SWPC code Np Slcr.

- **Database Lookup:** The generated SWPC codes are used to locate the corresponding Chinese compound names in the relevant database, using Unicode encoding for retrieval and representation.

While SWPC-SIFT approach effectively captures phonological features in a concise format, it currently operates at a semi-automated level, requiring further development for full automation. And achieving full automation will require intelligent matching methods and extensive training on large datasets. Despite this limitation, the approach fundamentally reflects our concept of how robots might perceive Chinese characters. By combining SWPC coding with multimodal recognition techniques for Chinese character images, the system can effectively recognize characters. This process demonstrates the potential of using SIFT and other machine learning technologies to identify radicals, components, as well as prefixes and roots of words, thereby achieving an initial level of intelligent character recognition.

# 5 CONCLUSIONS

This paper emphasizes the unique advantages of Chinese characters in multimodal language processing, particularly for humanoid robots. As a language system that intricately integrates pictographic and phonetic elements, Chinese characters offer distinct benefits in expressing complex scientific concepts and conveying cultural nuances. By examining the localization of the periodic table in Chinese and the naming conventions for organic compounds, this study demonstrates how the structural properties of Chinese characters open new opportunities for natural language processing (NLP).The main contributions of this paper are summarized as follows:

- Advocating for the adoption of the systematic approach observed in Chinese chemical nomenclature to advance research in Chinese natural language processing (CNLP).

- Demonstrating the efficacy of Six-Writings Pictophonetic Coding (SWPC) technology in encoding and recognizing Chinese characters by utilizing both visual and phonetic features.

- Proposing a novel multimodal processing framework that integrates SWPC with the Scale-Invariant Feature Transform (SIFT) method to enhance character recognition.

- Establishing the feasibility of leveraging Chinese characters as cross-lingual carriers in humanoid robot language processing.

As a cultural treasure and a linguistic heritage of Chinese civilization, Chinese characters have gained new significance in the era of artificial intelligence. By further exploring the phonosemantic encoding mechanisms and the hierarchical structure of Chinese character components, researchers can better harness the strengths of Chinese characters, thereby enhancing language comprehension and generation capabilities in humanoid robots. These developments will contribute to more intelligent and harmonious human-robot interactions in diverse linguistic and cultural contexts.

Future research should focus on two key areas: (1) enhancing the SWPC concept and expanding the SWPC library and image database to cover a broader range of Chinese characters, including both simplified and traditional forms, and (2) developing intelligent algorithms to facilitate the efficient search and combination of Chinese characters using radicals and components, as well as constructing words through affixes and roots for sentence-level analysis. These advancements will pave the way for more effective language processing in humanoid robots, enabling them to explore additional languages and create more sophisticated human-machine interaction scenarios.

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

# A  APPENDIX

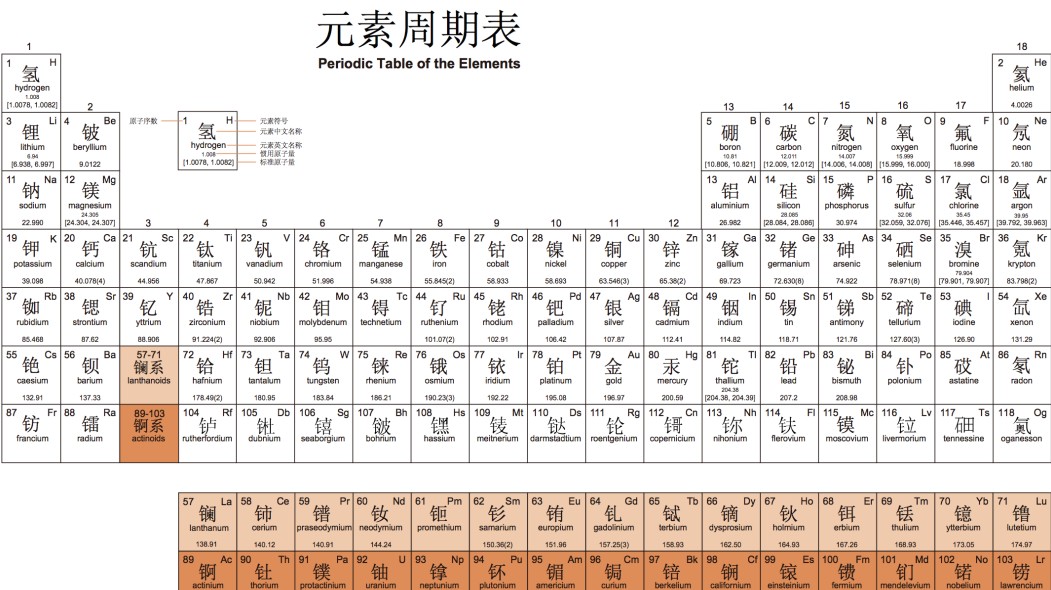

Figure 8: Chinese version of the periodic table, produced by the Chinese Chemical Society from the International Union of Pure and Applied Chemistry (IUPAC).

