# OpenReview forum: "Advancing Cross-Lingual Capabilities for Humanoid Robots: Leveraging Chinese NLP through Pictophonetic Advantages"
_ICLR.cc/2025/Conference — ICLR 2025 Conference Withdrawn Submission_

### Official Review · Reviewer_eWDF · 2024-10-30

**Soundness:** 1
**Presentation:** 2
**Contribution:** 1
**Rating:** 1
**Confidence:** 5

**Summary:**

This paper discusses the potential advantages of integrating Chinese natural language processing (CNLP) into humanoid robots to enhance their cross-lingual capabilities. The authors argue that Chinese characters, with their combination of ideographic, phonetic, and semantic components, offer a rich and multidimensional data source that can be leveraged for more efficient multilingual processing.

The paper references the Six-Writings Pictophonetic Coding (SWPC) framework, which enables the direct integration of Chinese NLP with morphological and semantic elements, using the Chinese version of the chemistry periodic table as an example. It goes on to discuss potential applications of SWPC to improve Chinese natural language understanding, such as:
- SWPC representation of Chinese characters and words
- Chinese character and word generation
- Chinese image/text recognition

However, no machine-learning-related experimentation or results are provided to support the arguments made in the paper, making it feel more like an introduction to existing methods and a proposal of how they might be applied rather than a complete study.

**Strengths:**

**Originality:**
- The paper introduces several ways in which the Six-Writings Pictophonetic Coding (SWPC) framework and the Scale-Invariant Feature Transform (SIFT) can be applied to improve Chinese natural language understanding and computer vision tasks, such as character identification.

**Clarity:**
- The paper provides concrete examples of Chinese characters in the periodic table of elements, demonstrating why the Six-Writings Pictophonetic Coding (SWPC) framework is a more compact and versatile representation of Chinese characters.
- Explanatory figures are used throughout the paper to enhance clarity.

**Significance:**
- The proposed methods have the potential to improve certain aspects of Chinese NLP tasks.

**Weaknesses:**

**Lack of Experimental Validation:**
- The paper does not provide any experimental results or quantitative analyses to substantiate the effectiveness of the proposed method.
- The only experiment mentioned in the paper is in section 4.3, where the author constructed an SWPC dataset of Chinese characters and image libraries, claiming that the application of SIFT helps in various contexts. However, no quantitative results are provided for this experiment.
- To improve the experiment, the author could select certain standard tasks in computer vision (CV) and natural language processing (NLP), define quantitative metrics such as accuracy, precision, recall, etc., and choose a few benchmark datasets to evaluate how the proposed methods perform in comparison to existing methods.
- Even if the method cannot outperform state-of-the-art deep-learning-based methods (as acknowledged by the author), demonstrating in which specific domain this method performs better and how its use improves interpretability would still be a valuable contribution. However, this requires quantitative evidence, i.e. comparative evaluations on benchmark datasets.

**Lack of Implementation Details:**
- The paper does not provide descriptions of how the proposed methods are implemented, making it difficult to reproduce and validate the results.

**Lack of Comparison:**
- The paper does not provide a quantitative evaluation against existing methods.
- A comparison is mentioned: "While some state-of-the-art (SOTA) deep learning algorithms, like MA-CRNN (Tong et al., 2020) and MaskOCR (Lyu et al., 2024), achieve higher accuracy in certain scenarios, SIFT’s interpretability and compatibility with SWPC make it an ideal choice for providing a comprehensive understanding of the processing and recognition of Chinese characters and words."
  - However, there are no descriptions of the specific scenarios in which each method performs better, and no analysis is provided to support the claims of interpretability and compatibility.

**Lack of Clarity:**
- The paper repeatedly mentions that the work is supposed to aid humanoid robots but does not discuss how. While the paper discusses various ways the methods can advance Chinese natural language processing, the only experiment presented is a computer vision task in section 4.3 on character/word recognition, which creates confusion about the scope and categorization of the paper.
- Figures 1 and 2 are low resolution.

**Questions:**

1. Can the authors provide quantitative experimental results to demonstrate the effectiveness of SWPC/SIFT compared to existing encoding methods in Chinese NLP?
2. How does SWPC/SIFT perform in terms of computational efficiency and accuracy in character recognition tasks compared to traditional and deep learning-based methods?
3. Why was SWPC/SIFT chosen over more recent deep learning approaches for character and word recognition? Can the authors justify this choice?
4. In what scenarios does SWPC/SIFT outperform state-of-the-art deep learning methods and in what domain it doesn't?
5. What experiments can be conducted to support the claim that SIFT methods offer better interpretability and compatibility?
6. Can the authors provide more detailed explanations or examples of how SWPC/SIFT integrates with machine learning models for language processing?

---

### Official Review · Reviewer_d4Wi · 2024-11-03

**Soundness:** 1
**Presentation:** 1
**Contribution:** 1
**Rating:** 1
**Confidence:** 3

**Summary:**

This paper explores the advancement in cross-lingual capabilities in humanoid robots by leveraging the structural richness of Chinese characters. Unlike phonetic languages, Chinese characters incorporate ideographic, phonetic, and semantic elements, making them information-dense and suitable for complex data representation. The authors introduce a novel Six-Writings Pictophonetic Coding (SWPC) system, which dynamically encodes Chinese characters. The paper does not provide evaluation of the method.

**Strengths:**

- Possibly a new idea to model Chinese character information.

**Weaknesses:**

- Lacks experimental evaluation
- No clear connection to humanoid robotics
- Inconsistent logical progression between the motivation and the proposed solution

**Questions:**

- What is the connection between the proposal and robotics?
- Could you provide any quantitative evaluation of this method?

**Details Of Ethics Concerns:**

I suspect this paper may have been generated by a Large Language Model (LLM). While the paper has a clear structure and syntactically correct sentences, there are inconsistencies in the narrative flow and poorly motivated discussions.

---

### Official Review · Reviewer_gibY · 2024-11-04

**Soundness:** 1
**Presentation:** 1
**Contribution:** 1
**Rating:** 1
**Confidence:** 4

**Summary:**

This work proposes using Chinese symbols to improve NLP in robotic systems. It focuses on periodic table symbols to achieve this and proposes a Six-Writing Pictophonectic Code. However, it does not provide evidence that this effort is being used with robots.

**Strengths:**

It is a good idea to reinforce the processing of Chinese symbols and all the types of information it conveys.

**Weaknesses:**

This work is minimal in several aspects that I will explain in the following points:

* The paper claims to be related to humanoid robots, but no evidence of a proposal on how to incorporate some aspects of the proposal is put forward.
* The case study is very narrow. It only focuses on periodic table symbols.
* The motivation for the case study is not clear.
* The evaluation is very narrow and associated with the case study, but no generalization is supported.
* The evaluation is qualitative and just exemplified; there is no fundamental notion of how good this approach would be and how it compares with other competing methods.

**Questions:**

* How would you evaluate the performance of the proposed method?
* How does this approach compare with other methods for Chinese symbol recognition?
* Why only consider the periodic table as a case study?
* Can there be other case studies that exemplify the proposed method?

---

### Note · Authors · 2024-11-13

I have read and agree with the venue's withdrawal policy on behalf of myself and my co-authors.